# Age-Related Changes in the Characteristics of the Elderly Females Using the Signal Features of an Earlobe Photoplethysmogram

**DOI:** 10.3390/s21237782

**Published:** 2021-11-23

**Authors:** Jeong-Woo Seo, Jungmi Choi, Kunho Lee, Jaeuk U. Kim

**Affiliations:** 1Digital Health Research Division, Korea Institute of Oriental Medicine, Daejeon 34504, Korea; jwseo02@kiom.re.kr; 2Human Anti-Aging Standards Research Institute, Uiryeong, Gyungnam 52151, Korea; jmchoi@brnd.co.kr; 3Gwangju Alzheimer’s Disease and Related Dementias (GARD) Cohort Research Center, Chosun University, Gwangju 61452, Korea; leekho@chosun.ac.kr; 4Department of Biomedical Science, Chosun University, Gwangju 61452, Korea; 5Dementia Research Group, Korea Brain Research Institute, Daegu 41602, Korea; 6Korean Convergence Medicine, University of Science and Technology, Daejeon 34054, Korea

**Keywords:** photoplethysmogram, Aging Index, third derivative PPG, feed forward neural network

## Abstract

Non-invasive measurement of physiological parameters and indicators, specifically among the elderly, is of utmost importance for personal health monitoring. In this study, we focused on photoplethysmography (PPG), and developed a regression model that calculates variables from the second (SDPPG) and third (TDPPG) derivatives of the PPG pulse that can observe the inflection point of the pulse wave measured by a wearable PPG device. The PPG pulse at the earlobe was measured for 3 min in 84 elderly Korean women (age: 71.19 ± 6.97 years old). Based on the PPG-based cardiovascular function, we derived additional variables from TDPPG, in addition to the aging variable to predict the age. The Aging Index (AI) from SDPPG and Sum of TDPPG variables were calculated in the second and third differential forms of PPG. The variables that significantly correlated with age were c/a, Tac, AI of SDPPG, sum of TDPPG, and correlation coefficient ‘r’ of the model. In multiple linear regression analysis, the *r* value of the model was 0.308, and that using deep learning on the model was 0.839. Moreover, the possibility of improving the accuracy of the model using supervised deep learning techniques, rather than the addition of datasets, was confirmed.

## 1. Introduction

Recently, the focus on healthcare for the elderly has shifted toward personal health monitoring using wearable devices. In particular, healthcare devices based on information and communication technology that can measure cardiovascular and musculoskeletal functions have been developed and commercialized as noninvasive wearable devices for daily health monitoring.

Monitoring the third derivative of the photoplethysmography (PPG) pulse (TDPPG) is a representative technology for daily monitoring of cardiovascular function. TDPPG represents the absorption of infrared light across a finger or earlobe, which has a pulsatile component and a constant component [1]. The TDPPG can indicate the cardiovascular flow, volume, and vessel-wall movement of blood, as well as the arrangement of the heart and arteries [2]. These cardiovascular functions are used for indirect evaluation of aging, hypertension, and diabetes.

The PPG pulse is used to measure the pulse-rate variability and to evaluate the sympathetic and autonomic nervous systems. Moreover, the values of various indicators related to the cardiovascular system can be determined through wave-morphology analysis. However, because recognizing the inflection point in pulses is challenging, the second derivative of the PPG pulse (SDPPG) and TDPPG are used for a detailed and objective evaluation. SDPPG is also known to be an indicator of arterial stiffness [3]. Generally, the inflection points based on the five upper and lower waves in SDPPG are calculated using the finger PPG. The first rising part indicates an increase in an early systolic pulse, the second indicates a decrease, the third denotes a systolic re-increase, the fourth represents a systolic re-decrease, and the final part indicates an early diastolic or diastolic. Cardiovascular characteristics based on systolic and diastolic pulses, which are difficult to identify accurately using PPG pulses, can be confirmed through differential form analysis [4]. Variables derived from SDPPG represent the cardiovascular function. For instance, the b/a ratio may reflect the large arterial stiffness. The b wave relative to the wave might be caused by decreasing distensibility of the aorta. There is a need to comprehensively identify and model representative variables that can confirm these cardiovascular functions.

The PPG pulse can be measured in various parts of the human body such as the finger, ear, wrist, and ankle. Generally, it is measured at the finger, and the result value measured at each position of the body shows different waveforms depending on the distance from the heart [5]. Since there are age characteristics, it is necessary to examine the characteristics of a specific age group and where to measure [6].

It is possible to develop indicators that can quantitatively evaluate the state of various physical functions in the elderly while measuring a PPG pulse; such indicators would reflect the cardiovascular function characteristics of the elderly [2]. In particular, it can be used as an indicator and criterion for cardiovascular function in the elderly with mild cognitive impairment or dementia. The evaluation criteria used for personal health data management for non-invasive measurements can be obtained by developing an indicator to confirm the vascular status with age [7]. The primary objective of this study is to quantitatively identify the levels of vascular aging of the elderly. AI based on SDPPG has been developed previously. For example, cardiovascular aging is estimated from SDPPG by leveraging the first inflection point of SDPPG and is obtained by dividing the difference between the remaining four inflection points [8]. A previous study developed a model that predicts the age of the elderly with high accuracy [3]. However, most previous methods predict aging based on the PPG pulse by AI from the SDPPG variable measured at the fingers.

In general, deep learning and machine learning techniques are applied to estimate age and vascular stiffness from measured PPG data. Some studies use the supervised machine learning to estimate ankle-brachial index from PPG [9]. In another study that extracted 38 features from PPG and estimated age by performing convolution neural networks and regression [10]. In addition, there are prior studies to acquire more accurate waveforms by deep learning the PPG signals measured in various parts of the body [11,12]. As such, deep learning and machine learning techniques are being used in various ways to improve the accuracy of waveforms or to make accurate estimations with acquired data.

Therefore, in this study, a new aging variable was derived using TDPPG, which can represent the inflection state of the pulse more precisely than SDPPG. In addition, we used linear regression with supervised learning deep networks and the common AI from SDPPG to obtain a more detailed age-related changes in the characteristics of the elderly females new aging variable. In addition, it is aim to check whether the age can be estimated from the results of measurements on the earlobe instead of the fingers.

## 2. Methods

### 2.1. Data Collection

The subjects of this study were 84 elderly women (age: 71.19 ± 6.97 years) with normal cognitive and musculoskeletal functions and no clinical findings of arrhythmias. Table 1 lists the characteristics of the participants. A consent form for the clinical trial was prepared based on the Institutional of Review Board (IRB, CNUH-2019-279) and approved prior to the experiment.

All subjects were asked to sit on a chair and maintain a stable state, and their heart rate pulse was measured using a PPG device (Model: EP520, LAXTHA Inc., Daejeon, Korea) attached to the left earlobe. The measurement duration was 5 min. The PPG sensor system used visible light with a wavelength of 640 nm and had a relative sensitivity of 60%. The sensor included two parts: a red LED as a transmitter and a photodiode as a receiver. The photodiode outputs current according to the intensity of the received light. The sensor circuit unit converts current into voltage, amplifies and filters the signal, and provides a final output signal. The frequency response of the receiver (photodiode) is 0.3–5 Hz in the −3 dB band in the characteristic curve (gain with frequency). This frequency response of the filter removes unwanted noise. The time delay of the output signal with respect to the input signal is 70.0 ms (Figure 1).

### 2.2. Data Analysis

The PPG data were measured at a sampling frequency of 250 Hz for 5 min. Detrending was performed preferentially in the initial preprocessing stage to remove trends and drifts. A fourth-order band-pass zero-phase Butterworth filter considering phase delay and distortion with a cut-off frequency of 0.4–9 Hz was used to remove low- and high-frequency noise from the DC components, such as movement noise and respiratory rhythm [13]. Morphologically, the point at which the notch was minimized after passing the dicrotic notch in the volume pulse trace was set as the end of the pulse. Just after is the start of the pulse, and defined as the pulse-to-pulse interval [14]. The representative pulse value was calculated as the average of each time point based on the start of the pulse, to obtain a representative single-pulse value from the values of the pulse-to-pulse interval. The second and third derivative pulses were used to obtain the SDPPG and TDPPG from each subject’s single-pulse PPG signal. Figure 2 shows the SDPPG and TDPPG signals in the form of the second and third derivatives in the continuous pulse data. The SDPPG comprises an a-wave (increase to systolic), b-wave (early systolic peak), c-wave (late systolic and dicrotic notch), d-wave (diastolic), and e-wave (decrease after pulse) [15]. MATLAB R2019a (MathWorks Inc., Natick, MA, USA) was used for signal processing and analysis of all experimental data.

#### 2.2.1. Multiple Linear Regression

Multiple linear regression analysis was performed to develop a model to predict the dependent variable, “age”. The independent variables were T_ab_, T_ac_, T_ad_, and T_ae_ (the time intervals between the waves) and b/a, c/a, d/a, and e/a (the a, b, c, d, and e-wave ratios of SDPPG); aging index (AI: (b-c-d-e)/a); vascular aging variable [8]; and sum of TDPPG (∑TDPPG(×10^4^)), which is the sum of the root mean square values of TDPPG. The regression analysis involved a stepwise method, and a model was developed with a dataset containing the information of 68 randomly selected subjects (approximately 80% of the 84 subjects). The test set comprised data from the remaining 16 subjects (corresponding to the remaining 20%), which were randomly extracted to verify the model.

#### 2.2.2. Fitting a Function with a Neural Network

We constructed a neural network to predict the age of an individual using the SDPPG and TDPPG variables. The c/a, T_ac_, AI, ∑TDPPG(×10^4^), four variables act as inputs to the neural network, and the real age is used as the target. A single hidden layer feed forward neural network (FFNN), with a comprising 15 neurons, was used for the neural network to learn age prediction (Figure 3). FFNN is a type of neural network that does not include connection loops between units. Information moves only in one direction from the input to the hidden layer; thus, no cycles or loops exist in the network. The error was calculated via back-propagation using the Levenberg–Marquardt weight. The FFNN model learned to minimize the error of the output value by iteratively updating the weights.

In this study, the input sample was divided into training, validation, and test sets. The training set was used to train the neural network, and the validation set was used from the training-set data until the desired accuracy was attained while training. The test set was divided based on a scale completely independent of the validation and training sets to verify the accuracy of the neural network. The test set provides a completely independent measure of network accuracy. The ratio of the test set is 20% of all data. The Levenberg–Marquardt method, which is a combination of the Gauss–Newton method and gradient descent method, was used as an optimization function when training the neural network model. It was used to measure the mean squared error (MSE), which was displayed on a logarithmic scale to evaluate the performance improvement of the neural network during training. The performance for each of the training, validation, and test sets was confirmed. Then, the network that exhibited the best performance on the validation set was selected as the final network. The MSE of the trained neural network was measured with respect to the testing samples. The regression was plotted across all data to check the fitting level of the final neural network [16].

## 3. Results

Owing to the correlation between age and other variables, the c/a, T_ac_, AI, and sum of the TDPPG variables were significantly correlated with age (Table 2). Figure 4 demonstrates the first-order linear fitting of the four correlated variables with age, c/a, T_ac_, AI, and the sum of TDPPG.

The age prediction model included only two variables. The AI and sum of TDPTG were selected based on the multiple linear regression analysis using the stepwise method, with four independent variables correlated with the dependent variable “age”. The adjusted-r^2^ value was 0.46 (*p* = 0.00). Table 3 presents the overall results of the model. The model developed using the predicted value is expressed using Equation (1): Figure 5 demonstrates the first-order linear fitting of the test set.
(1)$$\mathrm{Age}=109.65+\;\left(22.16\;\times \;\mathrm{AI}\right)\;-\;\left(0.05\;\times \;\sum \mathrm{TDPTG}\;\times \;10^{4}\right)$$

As a result of FFNN training, The MSE rapidly decreased as the network was trained; the performance is shown in Figure 6 for the training, validation, and test sets. The selected network performed the best on the validation set. The optimal performance of the training model, MSE, was 16.023, obtained for three of nine total epochs. The MSE was the lowest at epochs 3, but after that, MSE increased in validation set and test set.

Figure 7 presents the regression results for the training, validation, test, and all datasets. For the regression model derived using FFNN, the *r* values were 0.855, 0.701, 0.859 and 0.839 for the training data, validation data, test data, and all data, respectively. The error histogram shows the error for the estimated result. Additionally, Bland-Altman plots were used to compare the output values of the actual age and the learning model.

## 4. Discussion

The second derivative comprised of five inflection sections: a, b, c, d, and e [17]. These values correspond to the peak amplitudes, and the indicators for the ratio of amplitude inflection sections or duration were used as evaluation variables. In general, the a- and b-waves correspond to the early systolic component with minimal influence from the reflection wave, the c-wave corresponds to the mid-systolic component, the d-wave corresponds to the post-systolic component, and the e-wave corresponds to the position of the diastolic component after the dicrotic notch wave or the position of the diastolic component [18]. However, it is difficult to detect the dichroic notch and the subsequent diastolic components because the wave does not have an inflection point corresponding to the age or PPG measurement location [19]. Therefore, it can be seen that when the PPG data showing the position is delayed at a smooth differentiated wave pulse. In this study, the a- and b-waves corresponded to a position similar to that of the general early systolic component, but the d-and e-waves were detected after the diastolic component. This is because the measurement location was the earlobe, and the characteristic of an inflection point curve is that the subjects in their 70s and 80s are relatively elderly [19,20]. This can be interpreted as a result of the a, b, c, and d-waves corresponding to the systolic component and the e-wave corresponding to the diastolic component in the case of young adults, whose measurements are recorded at the fingertips, as mentioned previously. However, when a difference exists in the PPG wave shape, as observed in this study, the inflection points are similar to the a-and b-waves, but the c- and d-waves correspond to the dicrotic notch and diastolic component, respectively, not the mid- and post-systolic waves.

As described in previous studies, the AI obtained by dividing the sum of the c- and d-waves by the a-wave is the blood transition capacity of the peripheral vascular system [21]. The c-wave is high and the d-wave is low when the blood flow is appropriate (which is observed in the young adults), and the c-wave is low when the blood circulation is poor (as observed in the elderly).

The results of this study indicate a significant correlation between age and the considered variables, i.e., sum of TDPPG, c/a, and AI of SDPPG.

First, the c/a, the c-wave is a waveform produced by the influence of the b- and d-waves. Previous studies have shown that c/a reflects reduced arterial stiffness and that it decreases with age [8]. It has been shown previously that the c/a ratio reflects decreased arterial stiffness, and the c/a ratio decreases with age [14]. In this study, the insignificant b-wave was observed to affect the c-wave, and it reflected the aorta tensibility as the first vascular response upon blood ejection from the left ventricle. The b/a ratio indicates whether the aortic blood volume rapidly increases at cardiac output; this variable increases when the corresponding vascular load decreases, whereas it decreases in the case of a large arterial stiffness.

The value of AI is obtained by dividing the difference between the b, c, d, and e-waves by the a-wave and is known to reflect chronological age. Our findings agree well with previously reported results [8]. AI is also an indicator of the functional aging of blood vessels and severe arteriosclerosis; its value increases even when blood vessels age in normal people.

The c-wave at the third derivative is defined as the systolic component (p1), and the d-wave is the late systolic or early diastolic component (p2) [17,18]. In the case of the third derivative, an advantage is that the inflection point of the second derivative can be identified more clearly. The variables calculated from TDPPG and the evaluation of arterial stiffness from pulse waves have previously demonstrated that mental stress significantly affects arterial stiffness [17]. The sum of the TDPPG is expected to represent the variation in the entire pulse as the sum of the area of the wave at the center of the inflection point. Moreover, it is a quantitative value that can generally represent the elasticity of blood vessels corresponding to the overall blood flow.

A regression model was developed to predict real age using three variables with a high correlation. The c/a variable was removed based on the stepwise method, and only AI and the sum of the TDPPG were selected. The correlation coefficient *r* of the developed model was 0.68, and the adjusted- *r*^2^ value indicated a model accuracy of 0.46. The results obtained using the randomly extracted test set showed that *r* was slightly lowered to 0.556 (*r*^2^ = 0.309), as shown in Figure 3. This means that AI and the sum of the TDPPG variables are improved when used independently.

In addition, FFNN was used to improve the accuracy of the prediction model. This improved the *r* value to 0.859, which was higher than the value (0.556) determined via the multiple linear regression analysis of the test set. Thus, model accuracy can be improved without increasing the number of subjects or datasets.

The MSE for the validation-set showed a stable state after epochs 3, but in the case of validation-set and test-set, it showed a tendency to increase again after epochs 3. This is due to over-fitting that occurs because the learning model is complex or the number of partitioned test-set data that is small. In this study, it is expected that the number of data set used in the test-set is small. First, the number of neurons in the hidden layer was adjusted to simplify the learning model (Table 4) (Figure 8).

As a result, the test-set showed optimal performance and test-set fitting resulted in 15 neurons set in this study. In order to prevent overfitting, it is necessary to perform learning by adjusting the dividing ratio of training-set, validation-set, and test-set, or regularization through drop-out that can be supplemented. The drop-out method was not considered that because of the limitation of the input variables number. Based on the results of this study, additional work is needed to find a model with variance and bias based on the training-set results. This is a limitation of this study.

As a result of the error histogram according to the application of the model, it had normality around 0, but the maximum error was confirmed to be about ±8 to 10. This is thought to be the effect of data inaccuracy or unconsidered input variables rather than model inaccuracy. The correlation between the actual age and the predicted value confirmed through the Blond-Altman plot was confirmed. As a result, it can be seen that most of the data is located in the area corresponding to the 95% limit of agreement, but some values deviate from this. It is necessary to find additional data and an appropriate neural network model to confirm higher agreement between the methods.

In this study, we derived a significant new TDPPG variable in addition to the commonly used Aging Index variable. These variables can be used as an input to the model. The estimated age resulting from the model can be compared with real age. A standard in generalized people can be identified by estimating younger or higher than the estimated age. However, this study was developing a model by small number of data. Therefore, there is a limitation that generalization is difficult with a small number of subjects, and the purpose of this study is to confirm a method for developing a model. Another limitation of this study is that it does not directly confirm the function of the cardiovascular system. Furthermore, using the earlobe as the measurement location of the PPG is not the usual location. However, in this study, age could be estimated using the PPG of the earlobe. In future research, we plan to acquire more data for model development and compare the results from data obtained from various locations as well as from the earlobe.

## 5. Conclusions

In this study, the age-specific change of characteristics of elderly Korean women using the signal characteristics of an earlobe wearable photoplethysmometer was confirmed. The inflection point was analyzed using the third and second derivative values of the PPG pulse, and a new variable was derived from a combination of the inflection points. The possibility of improving the accuracy of the model using supervised deep learning techniques, rather than the addition of datasets, was confirmed. In future studies, additional data must be acquired to improve the accuracy of the model and to establish a physiological basis for these results.

## Figures and Tables

**Figure 1 sensors-21-07782-f001:**
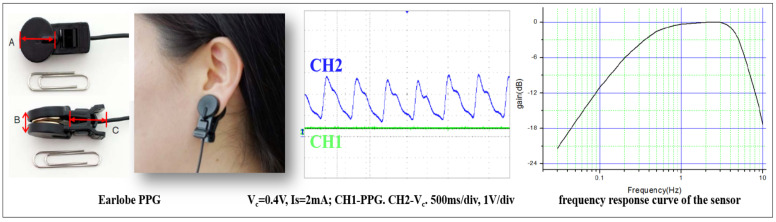
Wearable earlobe type PPG device and specification.

**Figure 2 sensors-21-07782-f002:**
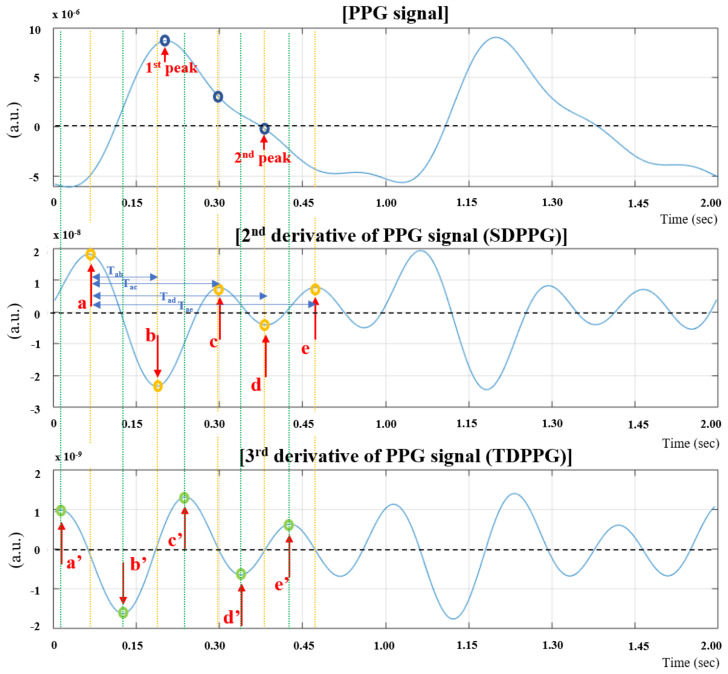
Description of PPG, SDPPG, and TDPPG indices.

**Figure 3 sensors-21-07782-f003:**
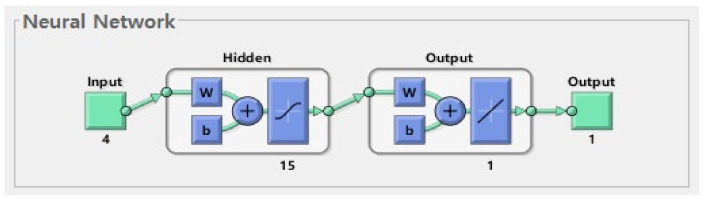
Structure of Neural Network.

**Figure 4 sensors-21-07782-f004:**
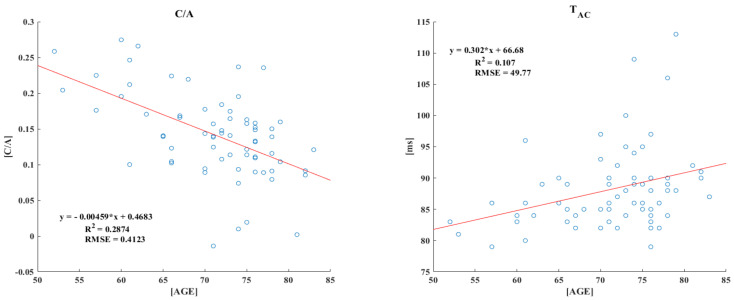
Scatter plot of the correlated variables with age.

**Figure 5 sensors-21-07782-f005:**
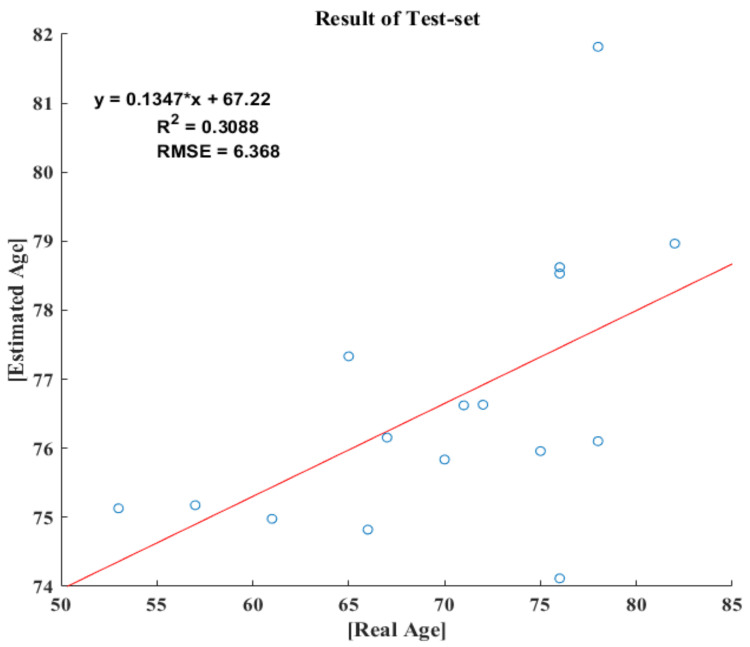
Results of predicted age by test-set.

**Figure 6 sensors-21-07782-f006:**
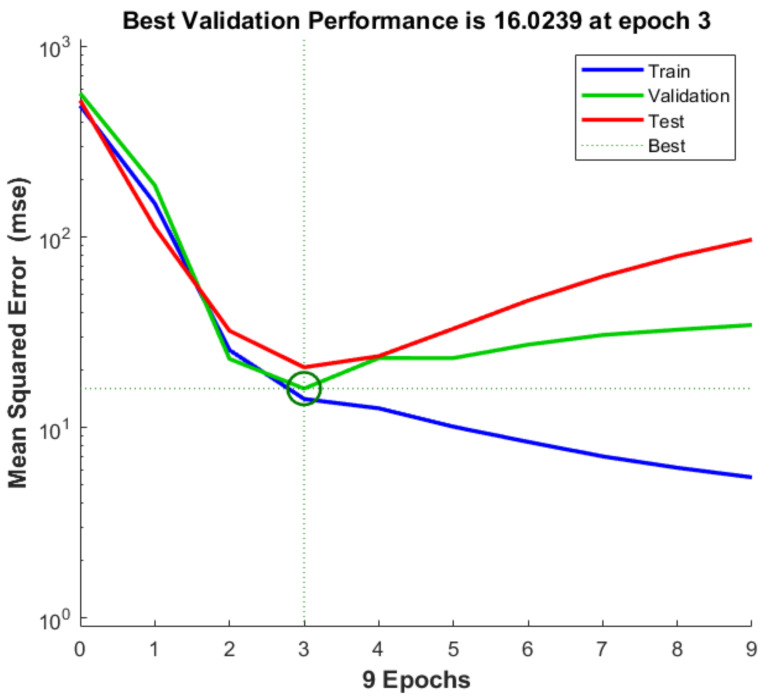
Result of training progress and best validation performance (15 hidden layer).

**Figure 7 sensors-21-07782-f007:**
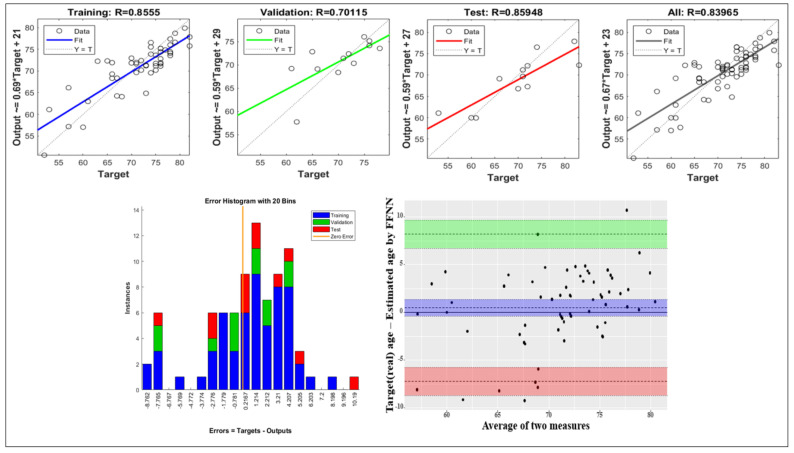
Result of neural network regression, Error histogram & Blond-Altman plot.

**Figure 8 sensors-21-07782-f008:**
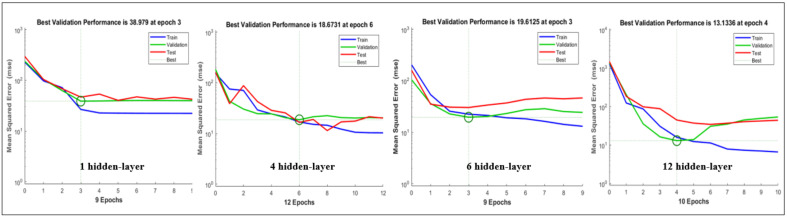
Result of MSE each hidden layer.

**Table 1 sensors-21-07782-t001:** Participant information.

Characteristic	Value
Age (year)	71.19 ± 6.97
Weight (Kg)	58.94 ± 9.33
Height (cm)	153.73 ± 8.11

**Table 2 sensors-21-07782-t002:** Results of the correlated variables with age.

	b/a	c/a	d/a	e/a	T_ab_ (ms)	T_ac_ (ms)	T_ad_ (ms)	AI	∑TDPTG × 10^4^
Mean ± SD	−1.10 ± 0.07	0.14 ± 0.06	0.03 ± 0.06	0.21 ± 0.05	44.21 ± 2.57	88.21 ± 6.48	108.07 ± 5.03	−1.49 ± 0.14	100.18 ± 0.05
r	0.31	−0.54	0.12	0.00	−0.05	0.33	0.08	0.53	−0.52
*p*	N.S	0.00 *	N.S	N.S	N.S	0.01 *	N.S	0.00 *	0.00 *

SD: Standard deviation, r: Pearson’s correlation coefficient (*: *p* < 0.05), N.S: Not significant.

**Table 3 sensors-21-07782-t003:** Results of multiple linear regression analysis.

	Estimate	Standard Error (SE)	t-Statistics (Estimate/SE)	*p*	Adjusted-r^2^
Intercept	109.65	6.56	16.73	0.00	0.46
AI	22.16	4.46	4.97	0.00	
∑TDPTG × 10^4^	−0.05	0.01	−4.81	0.00	

**Table 4 sensors-21-07782-t004:** Results according to the number of hidden layer.

No. of Hidden Layer	1	2	3	4	5	6	7	8	9	10	11	12	13	14	15	16	17	18	19	20
MSE	38.98	27.25	47.91	18.67	20.11	19.61	31.48	47.44	28.77	21.42	153.92	13.13	40.98	30.01	16.02	45.51	52.75	39.33	60.17	70.98
Best epoch	3	7	7	6	8	3	4	7	4	6	1	4	5	5	3	6	5	3	3	5
Training (r)	0.67	0.77	0.75	0.85	0.88	0.79	0.83	0.89	0.83	0.86	0.31	0.84	0.94	0.78	0.86	0.89	0.87	0.75	0.92	0.59
Validation (r)	0.55	0.71	0.74	0.75	0.41	0.76	0.51	0.65	0.68	0.79	0.39	0.91	0.78	0.43	0.71	0.64	0.23	0.58	0.58	0.21
Test (r)	0.72	0.35	0.11	0.63	0.48	0.17	0.69	0.50	0.77	0.17	0.82	0.64	0.74	0.52	0.85	0.75	0.73	0.78	0.78	0.28
All (r)	0.64	0.72	0.61	0.82	0.81	0.74	0.76	0.77	0.79	0.77	0.38	0.81	0.83	0.71	0.83	0.80	0.78	0.67	0.81	0.41

Best epoch: best validation performance epoch. r: Pearson’s correlation coefficient.

## Data Availability

The data presented in this study are available on request from the corresponding author.

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
