# Peer review of "Age-Related Changes in the Characteristics of the Elderly Females Using the Signal Features of an Earlobe Photoplethysmogram"

_sensors, 2021, doi:10.3390/s21237782_

Round 1

Reviewer 1 Report

The lines 175-180 and 208-216 need the corrections.

Author Response

Response to Reviewer 1 Comments

Dear Reviewer #.1,

Thank you for reviewing my manuscript. Also, thank you again for the advice.

We have made the following corrections as your review comments. We would appreciate it if you could review the author's comments again.

Point 1: The lines 175-180 and 208-216 need the corrections.

Response 1: Table 2 & 3, which corresponds to 175-180, 199-200 represents the “Results of the correlated variables with age” and “Results of multiple linear regression analysis”. According to your opinion, some variables shown in bold have been corrected, and line spacing has been adjusted to the form. Thank you.

Reviewer 2 Report

In this manuscript, the authors proposed to estimate the level of vascular aging based on PPG features. The PPG signals were measured using earlobe type sensor. I have the following critical concerns: - The authors’ definition on technical concepts such as neural network and regression are ambiguous and insufficient. It seems the authors have extracted several features (what the authors called independent variables) from the second derivative and the third derivative of PPG signals. However, no reason or any proof regarding the features were provided in the manuscript. - Different from the conventional approaches, the PPG signals have been measured using the earlobe type PPG sensor in the manuscript. No technical support on this selection has been provided in the manuscript. - Although an estimation of vascular aging levels is among the main objectives of this work, no discussion or technical proofs have been provided in the manuscript. - The authors’ analysis on their results are also insufficient and inaccurate. - The manuscript contains lots of grammar errors and non-logical descriptions.

Author Response

Response to Reviewer 2 Comments

Dear Reviewer #.2,

 Thank you for reviewing my manuscript. Also, thank you again for the advice.

We have made the following corrections as your review comments. We would appreciate it if you could review the author's comments again.

In this manuscript, the authors proposed to estimate the level of vascular aging based on PPG features. The PPG signals were measured using earlobe type sensor.

I have the following critical concerns: -

Point 1: The authors’ definition on technical concepts such as neural network and regression are ambiguous and insufficient.

Response 1: At the 2.2.2, the form of the neural network model fitting function with a neural network is shown in “figure 3”. Segmentation of the data in the analysis method was also considered, and this was additionally described. (line 178~179) Thank you.

Point 2: It seems the authors have extracted several features (what the authors called independent variables) from the second derivative and the third derivative of PPG signals. However, no reason or any proof regarding the features were provided in the manuscript.

Response 2: The rationale for the variables presented as independent variables was additionally described in introduction. In particular, the SDPTG and TDPTG variables referenced previous studies used to create age estimation models. These variables were confirmed that explain the cardiovascular function. (line 56~60) Thank you.

Point 3: Different from the conventional approaches, the PPG signals have been measured using the earlobe type PPG sensor in the manuscript.

Response 3: In general, the PPG signal is measured at the finger. However, there are previous studies that measure the position of the earlobe and toes as well. This has already been presented in the reference in the discussion. (ex. “This is because the location of the measurement is the earlobe, and the characteristic of an inflection point curve is that the subjects in their 70s and 80s are relatively elderly [20, 21]”) (line 270~272)

In addition, this study conducted to consider the wearable measuring device such as an EEG combined with PPG. analysis and results of the measured values. Please understand it. Thank you.

Point 4: No technical support on this selection has been provided in the manuscript.

Response 4: The explanation of data post-processing and variable extraction has already been presented in 2.2 data analysis. In order to provide technical information, the specification of the measuring device has been further added. Thank you.

Point 5: Although an estimation of vascular aging levels is among the main objectives of this work, no discussion or technical proofs have been provided in the manuscript.

Response 5: Since it is difficult to directly measure vascular aging, it is a common method to estimate vascular aging with indicators analyzed from PPG. Technical evidence for vascular aging was supplemented by references in the discussion by results (line 284~299). Thank you.

Point 6: The authors’ analysis on their results are also insufficient and inaccurate.

Response 6: Additional analysis has been added to check the accuracy of the model from the results derived through FFNN. It is common to review the performance of the model with MSE, and we added an analysis to check the performance by adding the results of adjusting the number of hidden-layers in the discussion. This is shown in <Table 4> and <Figure 8>. Thank you.

Point 7: The manuscript contains lots of grammar errors and non-logical descriptions.

Response 7: This manuscript has already been edited using English writing and editing experts (Editage Inc.). However, according to the point that you pointed out the need for additional grammar errors and non-logical descriptions manuscript, The English re-correction was made. The certification of this is attached below. Please let me know if there are any additional English errors. Thank you.

Reviewer 3 Report

The paper reports about the feasibility to estimate the age of individuals from PPG signals features. The paper is interesting, however some concerns need to be addressed:

MAJORS

  • In the introduction, a description of the background of the research is missing. Particularly, there are not references to works regarding the employment of machine learning and deep learning for estimating vascular age and stiffness from PPG signals. Please, provide a deeper description of this aspect. Please refer to:
    • Perpetuini, David, et al. "Photoplethysmographic prediction of the ankle-brachial pressure index through a machine learning approach." Applied Sciences6 (2020): 2137.
    • Dall’Olio, Lorenzo, et al. "Prediction of vascular aging based on smartphone acquired PPG signals." Scientific reports1 (2020): 1-10.
    • Chiarelli, Antonio M., et al. "Data-driven assessment of cardiovascular ageing through multisite photoplethysmography and electrocardiography." Medical engineering & physics73 (2019): 39-50.
    • Hsiu, Hsin, et al. "Discrimination of vascular aging using the arterial pulse spectrum and machine-learning analysis." Microvascular Research(2021): 104240.

  • The aim of the study was to predict the age of the participants from features of the PPG signals. In order to generalize the results, a wider range of ages should be considered. In this manuscript, participants’ age ranges from around 50 to 80 years (from figure 2). Why did the Authors choose this age range for the study? Are the results indeed generalizable for a wider age range? Please justify this aspect.

  • Figure 4 reports the mean square error in function of the epochs. It should be highlighted that the RMSE should reach a plateau after some epochs. In this case, it seems that the neural network finds a minimum local and then the error increases (the network goes in overfitting, hence the results are related to the sample and they are not generalizable). Some methods to reduce the overfitting have been proposed, such as the drop out and regularization. Please, provide more generalizable results.

  • Concerning the multiple linear regression, the Authors divided the study sample in train and test set, whereas, for the neural network, they divided the study sample in train, test and validation set. Why did the Authors choose two different approach? Please explain in the manuscript. Moreover, it is not clear whether the split of the study sample is performed once or it is repeated randomly with a bootstrap procedure, in order to provide a result that is indeed not dependent to the study sample.

MINORS

  • Please add the unit of measure in line 102

  • In order to show the performance of the model, please provide the Bland-Altman plot between the real and predicted age.

Author Response

Response to Reviewer 3 Comments

Dear Reviewer #.3,

Thank you for reviewing my manuscript. Also, thank you again for the advice.

We have made the following corrections as your review comments. We would appreciate it if you could review the author's comments again.

The paper reports about the feasibility to estimate the age of individuals from PPG signals features. The paper is interesting; however, some concerns need to be addressed:

MAJORS

Point 1: In the introduction, a description of the background of the research is missing. Particularly, there are not references to works regarding the employment of machine learning and deep learning for estimating vascular age and stiffness from PPG signals. Please, provide a deeper description of this aspect. Please refer to:

1) Perpetuini, David, et al. "Photoplethysmographic prediction of the ankle-brachial pressure index through a machine learning approach." Applied Sciences6 (2020): 2137.

2) Dall’Olio, Lorenzo, et al. "Prediction of vascular aging based on smartphone acquired PPG signals." Scientific reports1 (2020): 1-10.

3) Chiarelli, Antonio M., et al. "Data-driven assessment of cardiovascular ageing through multisite photoplethysmography and electrocardiography." Medical engineering & physics73 (2019): 39-50.

4) Hsiu, Hsin, et al. "Discrimination of vascular aging using the arterial pulse spectrum and machine-learning analysis." Microvascular Research(2021): 104240.

Response 1: According to the comment that there is no mention of research on the use of machine learning and deep learning for estimating blood vessel age and stiffness from PPG signals, it was additionally presented in the introduction that related prior studies were performed with reference to the four papers recommended. ([9,10,11,12]) (line 85-93) Thank you.

Point 2: The aim of the study was to predict the age of the participants from features of the PPG signals. In order to generalize the results, a wider range of ages should be considered. In this manuscript, participants’ age ranges from around 50 to 80 years (from figure 2). Why did the Authors choose this age range for the study? Are the results indeed generalizable for a wider age range? Please justify this aspect.

Response 2: This study was conducted to establish a comparative standard to determine the cardiovascular aging level of geriatric patients such as dementia and mild cognitive impairment. Therefore, the experiment was conducted based on the elderly aged 50 to 80 years. Regarding this in the introduction, Therefore, adding the sentence “In particular, it can be used as an indicator and criterion for cardiovascular function in the elderly with mild cognitive impairment or dementia in general.” at introduction part. (line 73-78) Thank you.

Point 3: Figure 4 reports the mean square error in function of the epochs. It should be highlighted that the RMSE should reach a plateau after some epochs. In this case, it seems that the neural network finds a minimum local and then the error increases (the network goes in overfitting, hence the results are related to the sample and they are not generalizable). Some methods to reduce the overfitting have been proposed, such as the drop out and regularization. Please, provide more generalizable results.

Response 3: As your comments, overfitting occurred after 3 epochs in the training data, validation data, and test data. This is an overfitting phenomenon that occurs because the model of the variables used for learning is complex or the number of divided test-set data not used for the training data is small. To prevent this, we know that it is necessary to perform learning by adjusting the dividing ratio of training data, validation data, and test data, or to regularize it through a drop-out method.

However, I do not think this method is appropriate because the number of variables used as input variables is just four. First, the number of hidden-layers was adjusted to reduce the complexity of the model. The results are presented in “Table 4”.

The overfitting problem was solved in the model with one hidden layer, which is the simplest structure, but the predictive power of the model is not high. Therefore, in order to solve this problem, various methods should be tried, and I think the most important is the method of adding a lot of data for model development. This was presented as a limitation of this study. We will try to overcome this through future research. Please understand it. Thank you. (line 333-347) Thank you.

Point 4: Concerning the multiple linear regression, the Authors divided the study sample in train and test set, whereas, for the neural network, they divided the study sample in train, test and validation set. Why did the Authors choose two different approach? Please explain in the manuscript. Moreover, it is not clear whether the split of the study sample is performed once or it is repeated randomly with a bootstrap procedure, in order to provide a result that is indeed not dependent to the study sample.

Response 4: The method used for multiple linear regression analysis was performed to confirm the variables selected when the optimal model was derived by the stepwise method. This is in 2.2.1. It was further mentioned in the method of multiple linear regression. (~ Multiple linear regression analysis was performed to develop a model to predict the dependent variable ~ (line 151-152)) In addition, in the discussion, the result of FFNN was compared with the r value, but the purpose of the discussion was not to directly compare the two methods, so the analysis method was performed differently. Thank you.

When dividing the samples, the test set provides a completely independent measure of network accuracy. The ratio of test set is 20% of all data. This is referred to as “2.2.2. It is additionally presented in Fitting a function with a neural network. (line 178-179)

MINORS

Point 5: Please add the unit of measure in line 102

Response 5: The measurement unit of PPG is voltage, and this is added at “2.1. Data Collection”. Thank you.

Point 6: In order to show the performance of the model, please provide the Bland-Altman plot between the real and predicted age.

Response 6: According to your comments, I added a Bland-Altman plot and error histogram to “figure 7”. Thank you.

Round 2

Reviewer 2 Report

First of all, my previous comments have not been addressed in the revised manuscript. Particularly, no reason for utilizing the earlobe type of PPG sensor is provided. This is critical because the third sentence of the second paragraph, page 2 of the revised manuscript stated the difficulty of sensing PPG with the earlobe type. Also, the last third sentence of the paragraph stated that the earlobe type with elderly people is challenging. Unfortunately, I could not find any supporting descriptions on this point in the revised manuscript.

Another critical concern is that the authors’ logic is not smooth over the entire manuscript. In the last sentence of the third paragraph, page 2 of the revised manuscript, the authors have mentioned that early works focused mostly on the aging prediction, not aging-level evaluation. However, what the authors have done in this work, as stated in the last sentence of page 2 of the revised manuscript, is also an aging index. In addition, no information or conclusion can be drawn from the fitting results shown in Fig. 5.

The manuscript also contains lots of flaws. The third sentence of Section 2.2.2 says that “Because the neural network starts with random initial weights, ~~, and a random seed was set to avoid this randomness.” Set a random seed to avoid randomness? I am not quite clear about what the authors want to claim in this paragraph. Moreover, their definitions of technical terminologies utilized in the manuscript are erroneous. For example, it is mentioned that two-layer feed forward NN is utilized, but I guess, what the authors have used is single hidden-layered feedforward NN (i.e., with a single hidden layer). Figure 3 is also inaccurate.

Finally, the revised manuscript contains lots of Grammar errors over the entire manuscript.

Author Response

Response to Reviewer 2 Comments

Dear Reviewer #.2,

 Thank you again for your interest and meaningful review. As you pointed out, many errors and corrections have been confirmed, and the corrections have been made by collecting opinions as much as possible. Once again, we ask for your meaningful review opinion. Modifications according to the comments of the second review are indicated in “blue” at the manuscript. Thanks a lot for your interest and advice.

Point 1: First of all, my previous comments have not been addressed in the revised manuscript. Particularly, no reason for utilizing the earlobe type of PPG sensor is provided. This is critical because the third sentence of the second paragraph, page 2 of the revised manuscript stated the difficulty of sensing PPG with the earlobe type. Also, the last third sentence of the paragraph stated that the earlobe type with elderly people is challenging. Unfortunately, I could not find any supporting descriptions on this point in the revised manuscript.

Response 1: Thanks for your meaningful comments. There was an error in writing. Therefore, the expression of meaning was not correct. What I wanted to mention in this paragraph is “The PPG pulse can be measured in various parts of the human body such as finger, ear, wrist and ankle. Generally, it is measured at the fingertip, and the result value measured at each position of the body shows different waveforms depending on the distance from the heart. Since there are age characteristics, it is necessary to examine the characteristics of a specific age group and where to measure.” Therefore, any incorrect expression sentence has been corrected. Thank you very much for your clear comments. (page2, line 61-65)

Point 2: Another critical concern is that the authors’ logic is not smooth over the entire manuscript. In the last sentence of the third paragraph, page 2 of the revised manuscript, the authors have mentioned that early works focused mostly on the aging prediction, not aging-level evaluation. However, what the authors have done in this work, as stated in the last sentence of page 2 of the revised manuscript, is also an aging index. In addition, no information or conclusion can be drawn from the fitting results shown in Fig. 5.

Response 2: Thanks for your meaningful comments. The purpose of this study is to explore the new aging index and finally predict cardiovascular function aging by synthesizing the existing AI and new variables. The title was first modified to suit the purpose of focused mostly on the “aging prediction”, not aging-level evaluation. And, we have excluded the use of the term “aging index”, which can be misleading. Also, the “aging index” of the revised manuscript was deleted as stated in the last sentence of page 2 of the revised manuscript and revised to “However, most previous methods predict aging based on the PPG pulse by AI from SDPPG variable measured at the fingers.”. In order to fit the purpose and results of this study, it has been explained according to the needs of this study. Sorry for suggesting that the logical part is not smooth. It has been modified as much as possible according to your comments. Thank you again for your understanding. (title, abstract, page 2 line73~79, 89, 91, 92, page 9 line284, page10 line346, 349)

Point 3: The manuscript also contains lots of flaws. The third sentence of Section 2.2.2 says that “Because the neural network starts with random initial weights, ~~, and a random seed was set to avoid this randomness.” Set a random seed to avoid randomness? I am not quite clear about what the authors want to claim in this paragraph.

Response 3: Thanks for your meaningful comments. There was an error in the expression of the sentence. This was a problem caused by directly citing the description of “MATLAB Software”, a tool for developing and analyzing this FFNN model. Random initial weights were not used in this study. I think it is right to delete the sentence as it may be misunderstood. We deeply apologize for any confusion. thank you.

Point 4: Moreover, their definitions of technical terminologies utilized in the manuscript are erroneous. For example, it is mentioned that two-layer feed-forward NN is utilized, but I guess, what the authors have used is single hidden-layered feedforward NN (i.e., with a single hidden layer). Figure 3 is also inaccurate.

Response 4: Thanks for your meaningful comments. At the sentence of “two-layer feed-forward NN”, the “two-layer” means a single hidden layer and an output layer. As your comment, I also think the definition of the term is wrong. The sentence may cause misunderstanding of the reader. So, it was corrected to “single hidden layer composed of 15 neurons”. We deeply apologize for any confusion. thank you.

Point 5: Finally, the revised manuscript contains lots of Grammar errors over the entire manuscript

Response 5: Grammar errors of “points 1-4” have been corrected. In addition to this, if there are any parts that need additional correction, please confirm. Once again, thank you for your kind comments, interest, and comments.

Reviewer 3 Report

The paper is considerably improved and, in my opinion, it is suitable for publication in the present form.

Author Response

Response to Reviewer 3 Comments

Dear Reviewer #.3,

 Thank you again for your interest and meaningful review. As you pointed out, many errors and corrections have been confirmed, and the corrections have been made by collecting opinions as much as possible. Once again, we ask for your meaningful review opinion. Thanks a lot for your interest and advice.

Point 1: The paper is considerably improved and, in my opinion, it is suitable for publication in the present form.

Response 1: Thanks for your meaningful comments.

Round 3

Author Response

The advantage of this study is two.

First, it is possible to derive chronological age by using the previously derived variable TDPPG variable together with the variable known as the aging index. It was also advanced using FFNN.
Second, this study checks whether age can be estimated from the results of PPG measurements on the earlobe instead of the fingers.
Thus, the last sentence of the introduction has been modified for the purpose of this study.

 Somebody can check Age, which is the "Y" value that is input to the developed model using the variables “Aging index” and “sum of TDPPG” in this study. Estimated age is higher or lower than real age to indirectly determine the level of cardiovascular function.

 The reason why it is said to be a function of the cardiovascular system is that PPG is an indicator of vascular elasticity and cardiovascular function. Of course, in order to generalize, we need to accumulate more data and increase the accuracy. This is a limitation of this study and was added at the end of the discussion.

 Let's take a similar previous study as an example. Correlation between age and d/a was also confirmed in a previous study by “Norili Inoue” in 2016 “Second derivative of the finger photoplethysmogram and cardiovascular mortality in middle-aged and elderly Japanese women”, which is an index confirming the function of cardiovascular function with age. 

“Kenji Takazawa” also developed an aging index in a previous study called “Assessment of Vasoactive Agents and Vascular Aging by the Second Derivative of Photoplethysmogram Waveform” and saw a correlation with age.
I think that the same work as this study work or deriving new variables and using the FFNN technique is another methodological result.

Most of the studies that estimate aging is making models by estimating observes age or chronological age, which is the "Y" value. A further example is “A novel strategy for forensic age prediction by DNA methylation and support vector regression model” in a Scientific report (2015). The author created a model for estimating chronological real age with variables derived from the DNA-methylation technique. Age being the Y value is a limit that occurs because the generally accepted golden standard is real (chronological) age. As you said, there are healthy people and unhealthy people. I don't think it's wrong to consider and model the various cases without taking these controls into account. Most aging or body age estimation studies use the same method as my research.

It is understandable that the use of age "Y", which is not a direct indicator for predicting cardiovascular function and is not suitable for the title of this paper. Therefore, we would like to change the title to “Age-related changes in the characteristics of the elderly females using the signal features of an earlobe photoplethysmogram”.

This manuscript was written through the methods and references of various previous studies. Please consider the methodological problems once again. 
